# Matrix Metalloproteinases in the Periodontium—Vital in Tissue Turnover and Unfortunate in Periodontitis

**DOI:** 10.3390/ijms25052763

**Published:** 2024-02-27

**Authors:** Dominik Radzki, Alessandro Negri, Aida Kusiak, Michał Obuchowski

**Affiliations:** 1Department of Periodontology and Oral Mucosa Diseases, Faculty of Medicine, Medical University of Gdańsk, 80-208 Gdańsk, Poland; 2Division of Molecular Bacteriology, Institute of Medical Biotechnology and Experimental Oncology, Intercollegiate Faculty of Biotechnology, Medical University of Gdańsk, 80-211 Gdańsk, Poland

**Keywords:** extracellular matrix, matrix metalloproteinases, periodontal diseases, periodontitis

## Abstract

The extracellular matrix (ECM) is a complex non-cellular three-dimensional macromolecular network present within all tissues and organs, forming the foundation on which cells sit, and composed of proteins (such as collagen), glycosaminoglycans, proteoglycans, minerals, and water. The ECM provides a fundamental framework for the cellular constituents of tissue and biochemical support to surrounding cells. The ECM is a highly dynamic structure that is constantly being remodeled. Matrix metalloproteinases (MMPs) are among the most important proteolytic enzymes of the ECM and are capable of degrading all ECM molecules. MMPs play a relevant role in physiological as well as pathological processes; MMPs participate in embryogenesis, morphogenesis, wound healing, and tissue remodeling, and therefore, their impaired activity may result in several problems. MMP activity is also associated with chronic inflammation, tissue breakdown, fibrosis, and cancer invasion and metastasis. The periodontium is a unique anatomical site, composed of a variety of connective tissues, created by the ECM. During periodontitis, a chronic inflammation affecting the periodontium, increased presence and activity of MMPs is observed, resulting in irreversible losses of periodontal tissues. MMP expression and activity may be controlled in various ways, one of which is the inhibition of their activity by an endogenous group of tissue inhibitors of metalloproteinases (TIMPs), as well as reversion-inducing cysteine-rich protein with Kazal motifs (RECK).

## 1. Introduction

Periodontitis is a common chronic inflammation, affecting the periodontium, tissues surrounding the teeth, composed of proteins and non-proteinaceous components, which altogether are referred to as the extracellular matrix (ECM). The pathogenesis is multifactorial, based on a complex interaction of local, immunological, genetic, and environmental factors [1]. Periodontitis results from the disturbed balance between the host and resident microbiome, leading to microbiome dysbiosis, which contributes to the host’s up-regulated inflammatory response, responsible for the destruction of the periodontal ECM [2]. Degradation of the ECM is associated with the activation cascade of matrix metalloproteinases (MMPs), which is a key feature in the pathogenesis of periodontitis [3]. During homeostasis, the production and activity of MMPs are finely controlled, regulating ECM turnover. However, during periodontitis, MMP activity is up-regulated to an extent far exceeding the regulatory control. The ongoing inflammation results in irreversible tissue loss, and subsequently in teeth loss, affecting overall health and well-being.

MMPs are among the most important proteolytic enzymes of the ECM [4]. Capable of degrading all proteinaceous ECM components, MMPs are involved in multiple processes in the body, playing a role not only in pathological but also in physiological processes [5]. MMP activity is implicated in various pathologies of the oral cavity, for example, dental erosion, dental caries, pulpitis, periapical periodontitis, and oral cancer invasion and metastasis [6,7]. Moreover, being implicated in odontogenesis and tissue turnover, impaired MMP activity may result in deficient enamel mineralization or in gingival overgrowth [8,9]. For such reasons, MMPs in periodontitis have received great popularity, although to a lesser extent for physiological processes.

The purpose of this review is to provide a profound insight into the molecular aspects of periodontal anatomy and how it is impaired during periodontitis. By describing the ECM and MMPs, and their interactions in development, hemostasis and diseases, affecting not only the periodontium but the entire body and oral cavity, we would like to provide an extensive scaffold for a better understanding of the role of MMPs in the homeostasis of the periodontium as well as in periodontitis. We also would like to highlight the involvement of bacterial proteases in the destruction of the periodontium. Furthermore, based on the notions described in this review, it will be also shown how MMPs and bacterial proteases represent a good therapeutic target in periodontitis therapy.

## 2. The Extracellular Matrix

The ECM (also matrisome) is a complex non-cellular three-dimensional macromolecular network present within all tissues and organs, forming the foundation on which cells sit. It is composed of 1. structural and specialized proteins (such as collagens, elastin, fibrillin, laminin, fibronectin, vitronectin, nidogen also known as entactin, tenascin, amelogenins, dentin sialoprotein, and dentin phosphoprotein), 2. glycosaminoglycans, also known as mucopolysaccharides (hyaluronan, heparin, heparan sulphate, chondroitin sulphate, keratan sulphate, and dermatan sulphate), which are the major components of proteoglycans (proteins that contain covalently linked glycosaminoglycans, such as syndecan, betaglycan, perlecan, aggrecan, decorin, andversican), 3. minerals (hydroxyapatite and derivatives found in bone or dentine), 4. lipids, 5. water, and 6. tissue-bound growth factors [10,11,12,13,14,15,16]. The most abundant proteins of the ECM are collagens [17,18]. The ECM in mammals is composed of around 300 proteins [19]. The ECM composition is unique in the tissue in which it is located and its roles and effects are also various. The ECM is not only tissue-specific but also heterogeneous [20]. The individual components of the ECM do not exist in isolation; they are capable of interacting with each other to contribute to the formation and stabilization of a comprehensive network [15].

The ECM plays vital roles in animal development and throughout life, providing a fundamental framework for the cellular constituents of tissue and biochemical support to surrounding cells [13,21,22]. The ECM (both its composition and mechanical properties) affect many cell functions, such as anchorage, morphogenesis, differentiation, homeostasis, signaling, and survival [20,23]. The ECM provides mechanical properties to each organ by its tensile, compressive strength and elasticity and maintains water retention [20]. The structure of the ECM has profound effects on the function of the tissue, providing additional properties to the ones conferred by the cellular components of the tissue alone [24]. The importance of the ECM is vividly illustrated by the wide range of syndromes, from minor to severe, that arise from genetic abnormalities affecting ECM proteins [20].

The ECM is divided into two types: the interstitial matrix (IM, also known as stromal matrix, or just ECM) and basement membrane (BM); the BM is a layer of the ECM in contact with all epithelia and endothelia, which separate tissues within the body, while the IM includes the ECM surrounding the cells in tissues [23,25].

The IM is a space surrounding cells of the dermis, lamina propria, glia and neurons, connective tissues and others [26,27,28]. The IM is secreted by cells such as fibroblasts, myofibroblasts, smooth muscle cells, glial cells and neurons, and others [26,29]. The composition is dynamic and unique in the tissue in which it is located. The main constituents of the IM are collagen, elastin, fibronectin, hyaluronan, and decorin [29,30]. The principal fibrous protein of the IM is type I collagen, with various amounts of other collagen types [23,31]. Elastin is another abundant fibrous protein. In contrast to collagen, there appears to be only one gene encoding elastin; nevertheless, variants arise by alternative splicing [32]. Collagens provide tensile strength to tissues, regulate cell adhesion, tissue development, and homeostasis [13,29,30]; meanwhile, elastic fibers play an important role in rendering tissues pliable to mechanical stresses of distension, bending, and twisting [15,33]. Hyaluronan, which does not contain a core protein, is another abundant IM component which stabilizes ECM integrity, regulating the hydration of tissues, as well as cell adhesion, migration, and mitosis [29]. The IM provides a fundamental scaffold in the body.

The BM is a thin but dense, specialized type of the ECM, always associated with cells, that separates the epithelium, endothelium, muscle fiber, and nervous tissue from any surrounding stroma (connective tissues) [34,35]. The BM has supporting functions and it is produced by epithelial cells (that line the outer surfaces of organs and blood vessels), endothelial cells (that line the interior surface of blood vessels and lymphatic vessels), and some mesenchymal cells [29,36,37,38]. The BM is a multilayered structure composed of type IV collagen, laminins, nidogen, perlecan, and many others [23,39,40,41,42]; the composition is diverse and dynamic [27]. The network of type IV collagen provides the tensile strength of the BM [43]; laminin plays a vital role in BM assembly, binding with epithelial cells and forming with type IV collagen two sheet-like networks [29]. According to electron microscopy, the BM consists of successive three layers: lamina lucida, lamina densa, and lamina reticularis (Figure 1). In brief, the lamina lucida (syn. lamina rara) is composed of laminin, containing anchoring transmembrane proteins (collagen type XVII and integrins) which connect the BM and hemidesmosomes (matrix–cell attachment structures), the lamina densa is composed mainly of collagen IV, connected with lamina lucida by laminins and by anchoring proteins (collagen VII and fibrillin) with the lamina reticularis (syn. lamina fibroreticularis), composed of reticular fibers (collagen I and III) [34,44,45,46]. Generally, there is terminological confusion with the BM-associated terms and their meaning: basal lamina and BM are used interchangeably, when other times basal lamina is rather a part of the BM, consisting of lamina lucida and lamina densa. Also, the BM is not a real membrane like the cell or nucleus membrane. The BM is more dense and less porous than the IM [30]. The BM serves as a physical selective barrier, divides tissue into compartments, and acts as a signaling platform; it takes part in cell anchorage, survival, differentiation, polarity, proliferation, and migration [27,47,48].

The ECM is a highly dynamic structure which is constantly being processed during physiological conditions (known as turnover), where old proteins are degraded and new proteins formed, or development and wound healing (known as remodeling) [19,20,49,50]. Dysregulation of the ECM, like disintegration, and excessive or uncontrolled remodeling, which alters composition, structure, stiffness, and abundance, contribute to several pathological conditions [19,49]. The ECM undergoes remodeling, either enzymatic and non-enzymatic, and its molecular components are subjected to countless post-translational modifications. The turnover is mediated by enzymes that are responsible for ECM degradation, such as adamalysins, meprins, and matrix metalloproteinases (MMPs), but also by factors inducing ECM synthesis, such as growth factors (GFs), of which the ECM is a reservoir [19,20,30]. MMPs are the key proteases of the ECM [30].

## 3. Matrix Metalloproteinases

MMPs (also known as matrixins) are zinc- and calcium-dependent endopeptidases, degrading collectively all proteinaceous ECM components through cleavage of internal peptide bonds [51,52]. They are among the most important proteolytic enzymes (proteases) of the ECM [4]. MMPs, through impact on the ECM, are involved in adhesion, survival, proliferation and differentiation, migration, and intercellular interactions [53]. MMPs or homologues are present in a wide range of organisms, such as viruses, some bacteria, plants, and animals [5,54,55,56]. Also, studies have identified potential MMPs in fungi [57]. 

### 3.1. Types of MMPs

The MMP family in humans has 23 members (or 24, including two equivalent forms of MMP-23, MMP-23A and MMP-23B, encoded by two distinct genes, despite 29 having been used in the literature) which can be distinguished in two main types: MMPs secreted to the ECM in the latent proenzyme form (proMMPs) and then activated extracellularly (except MMP-11, MMP-21, and MMP-28), and membrane-type metalloproteinases (MT-MMPs), which undergo processing in the cellular compartment and are subsequently attached to the cell membrane in the activated form [52,58]. Historically, they were classified into six groups based on substrate specificity or homology: collagenases, gelatinases, stromelysins, matrilysins, MT-MMPs, and other MMPs (Table 1). Despite this classification, most MMPs can degrade several substrates with different specificities; for example, gelatinases can degrade several collagen types, and collagenases also degrade gelatin, although the rate of this degradation is much slower than that of gelatinases [59]. Therefore, MMPs are also classified into eight groups according to their structure (Table 2) [60,61]. Different MMPs may cooperate in order to degrade a protein substrate completely [62]. MMPs can also modify several non-matrix proteins such as cytokines, chemokines, growth factors, and adhesion molecules, suggesting that MMPs may have complex regulatory roles over various chemical mediators [52,63]. Generally, they act in neutral pH.

### 3.2. MMP Structure

MMPs together with astacins, adamalysins (a disintegrin and metalloproteinase, ADAMs, and ADAMs with thrombospondin motifs, ADAMTSs), snake venom proteinases, serralysins, meprins, leishmanolysins, and pappalysins belong to the metzincin superfamily (clan), named after a specialized structural component. It consists of a zinc-binding conserved sequence HExxHxxGxxH/D (‘x’ is any amino acid residue) within their catalytic domain and an invariant methionine-containing 1,4-β-turn called Met-turn [5,52,56,62,66,67]. Metzincins are present in all of the kingdoms of life [56].

MMPs are multidomain proteins, generally composed of 1. a signal N-terminal peptide, known as a predomain, of inconsistent lengths, controlling peptide secretion and being removed in the endoplasmic reticulum, 2. a prodomain, of about 80 amino acid residues, which keeps MMPs inactive until removed, 3. a catalytic zinc-dependent domain of about 160-170 residues, 4. a flexible proline-rich linker peptide, known as a hinge region, of inconsistent lengths and about 14-69 amino acids, 5. a hemopexin-like domain of about 200-210 residues, and 6. an additional transmembrane domain, present in some MMPs only [5,21] (Figure 2).

The prodomain of MMPs has a cysteine switch motif PRCGxPD, which is a highly conserved region; the cysteine sulfhydryl group in that motif chelates the catalytic zinc in the active site, keeping MMPs in their inactive zymogen form (proMMPs) until cleaved by protease [62]. Within the active site of the catalytic domain, a zinc ion, bounded by three histidine residues present in the conserved sequence HExxHxxGxxH (a zinc-binding motif), and three calcium ions are found [62]; glutamic acid adjacent to the first histidine residue is essential for the catalytic process [68]. The Ω-loop found in the catalytic domain, with significantly varying length and amino acid composition among MMPs, is responsible for different selectivity [5,69]. In the terminal zone of the catalytic domain, located eight residues down from the third histidine residue in the zinc-binding motif, there is a conserved loop—the met-turn—which is present in MMPs as well as in other metzincin families [56,62,70]. A second zinc ion found in the catalytic domain is necessary only for a structural purpose—maintaining protein conformation [5]. The linker (the hinge) is a proline-rich region between the catalytic and hemopexin-like domains responsible for inter-domain flexibility and enzyme stability, influencing enzyme collagenolytic activity. Indeed, mutations in the linker region, which affects inter-domain flexibility, reduce its ability to hydrolyze structurally complex substrates [5,71]. The hemopexin-like domain is important for substrate specificity and crucial for collagen triple-helix degradation, whereas the catalytic domain is sufficient for the degradation of non-collagen substrates [5]; the flexibility is fundamental for the interaction between the hemopexin-like and catalytic domains around the collagen triple helix, resulting in the bending of the collagen molecule, allowing for the unfolding of a single collagen strand and exposing it to the catalytic domain active site, enabling its subsequent hydrolysis [72,73]; the domain is present in all MMPs except MMP-7 and MMP-26 (matrilysins), or in the case of MMP-23, substituted by an another domain (an immunoglobulin-like domain and a cysteine-rich domain) [5,74].

### 3.3. Expression and Activity Regulation

MMPs are widely produced by multiple tissues; they are generated by connective tissues and proinflammatory cells such as fibroblasts, osteoblasts, endothelial cells, macrophages, neutrophils, and lymphocytes under the control of various substances [4]. MMPs are regulated at various levels including gene transcription, mRNA translation, protein compartmentalization, activation of the proenzyme forms as well as inactivation of the enzyme—by the counteracting actions of antiproteases. Some MMPs coding genes are not constitutively expressed by cells. Their production is induced and regulated by hormones, growth factors, proinflammatory cytokines, epigenetic modifications such as DNA methylation and histone modifications, as well as following changes in cell–matrix and cell–cell interactions [5,21]. Proinflammatory cytokines (such as TNF-α or IL-1β) activate MMPs genes via binding transcription factors (e.g., activator protein-1 and mitogen-activated protein kinase) [21]. MMPs are synthesized as inactive pre-proMMPs, from which the signal peptide (predomain) is removed during translation to generate proMMPs (zymogens, with the prodomain) [62]; subsequently, a stepwise activation of the latent zymogens through proteolytic removal of the prodomain (with the cysteine switch) takes place in the ECM, except for MT-MMPs [75]. MT-MMPs have a furin-like proprotein convertase recognition sequence at the C-terminus of the propeptide, allowing for its intracellular activation by furin; also MMP-11, MMP-21, and MMP-28 containing a furin cleavage site are activated intracellularly by furin [62]. ProMMPs can be activated by endopeptidases such as serine proteases (such as urokinase-type plasminogen activator, uPA, tissue plasminogen activator, tPA, plasmin, furin, or thrombin) and even by other active MMPs or microbial proteinases [58,62,76,77,78]. MMP activation may also be a result of physicochemical agents, such as low pH and heat, as well as chaotropic agents and thiol-modifying agents, such as 4-aminophenylmercuric acetate, mercury chloride and N-ethylmaleimide, oxidized glutathione, and sodium dodecyl sulphate, causing disruption of the cysteine–zinc ion coordination at the cysteine switch motif of the MMP [5,62].

The enzymatic activity of MMPs is down-regulated by endogenous polypeptides—tissue inhibitors (antiproteases) of MMPs (tissue inhibitor of matrix metalloproteinases, TIMPs) [79]. TIMPs (TIMP-1, TIMP-2, TIMP-3, and TIMP-4) inhibit MMPs through reversible blockage, forming 1:1 stoichiometric complexes; TIMPs selectively inhibit different MMPs as well as members of the ADAM and ADAMTS families. TIMPs have a low specificity for MMPs and each TIMP can inhibit several MMPs with a different efficacy [62]. The ability of TIMPs to inhibit MMPs is a result of interaction between the TIMP terminal N-domain and MMP catalytic domain, where the C-domain is involved in the interaction with the hemopexin domain of some MMPs [60]. Whereas TIMPs are the primary inhibitors of MMPs in tissues, α2-macroglobulin is the primary regulator of MMP activity in body fluids (another is α1-antitrypsin); α2-macroglobulin is a non-specific, broad spectrum inhibitor which inhibits most active proteases by trapping them as a result of proteolysis of α2-macroglobulin’s bait region by a protease, which results in a conformational change into a tetrameric cage around active proteases [21,59,80,81]. Another inhibitor is RECK (abbreviation of reversion-inducing cysteine-rich protein with Kazal motifs), an important mediator of tissue remodeling, able to inhibit MMP-2, MMP-7, MMP-9, and MT1-MMP [82,83]. However, in contrast to what has been previously reported, nowadays RECK is considered not to be a direct inhibitor of MMP catalytic activity, but to be able to regulate MMPs through other mechanisms, such as by down-regulating transcription or translation, or reducing the levels of extracellular MMPs by limiting their secretion or by binding and sequestering them [83,84]. Apart from the natural inhibitors of MMPs, several pharmacological inhibitors of MMPs have been found and employed in the treatment of some diseases.

While TIMPs inhibit MMPs by binding to the active site of the enzymes, they can also participate in the activation of MMPs. TIMPs form a non-covalent complex with proMMPs by interacting with the hemopexin domain via the C-terminal domain, acting as a linker between MT-MMPs and proMMPs at the cell surface [85]. This process is only claimed for TIMP-2 [86]; according to the current paradigm, TIMP-2 first forms a complex with proMMP-2 by binding to its hemopexin domain, after which the complex localizes onto the cell surface where it binds to the active site of an MT1-MMP molecule, forming a ternary complex which permits the activation of proMMP-2 by another free MT1-MMP [85,87]. TIMP-4 is also able to form a complex with MT1-MMP and the hemopexin domain of MMP-2, but does not lead to their activation [86]. Other non-inhibitory MMP-TIMP complexes have been shown [88]. It is suggested that understanding the structure of the non-inhibitory TIMP-MMP complexes should significantly aid in the design of highly specific inhibitors and optimize therapeutic procedures in MMP-dependent diseases [89].

## 4. A Brief Glance at the Role of Matrix Metalloproteinases in the Development and Diseases of the Oral Cavity

MMPs are involved in the development and diseases of the oral cavity as well; a schematic summary is presented in Figure 3.

MMPs are implicated in odontogenesis, playing a role in morphogenesis and cytodifferentiation; they are involved in the formation of enamel and dentine. It has been shown that MMP-3-induced degradation of proteoglycan in predentin near the mineralization front is a prerequisite for hydroxyapatite formation [90], when MMP-20 (enamelysin) is mainly expressed in ameloblasts, cleaves amelogenin, and regulates enamel mineralization [91]. MMPs are also implicated in epithelial–mesenchymal interactions during the initial stages of odontogenesis [90]. The lack of degradation of amelogenin, ameloblastin, enamelin as well as dentin sialophosphoprotein (DSPP), which is a precursor for dentin sialoprotein (DSP), dentin glycoprotein (DGP), and dentin phosphoprotein (DPP), and their subsequent accumulation in the enamel may be responsible for deficient enamel mineralization, and that may result in amelogenesis imperfecta or be associated with molar–incisor hypomineralisation [90,92]. MMPs also play a significant role in the processing of dentin sialoprotein, which is essential for dentinogenesis; the absence of MMP-9 may display a phenotype similar to dentinogenesis imperfecta, including decreased dentin mineral density and abnormal dentin architecture [64]. MMPs are also involved in processes in mature dentine. MMP-3 accelerates angiogenesis during pulp wound healing; it is essential for the formation of tertiary dentin, which plays a role as a barrier against bacterial, chemical, and physical stimuli [93,94].

In addition, there are plenty of oral non-developmental problems associated with MMP production and activity. Dental caries is a prominent multifactorial condition affecting the teeth, caused directly by microorganisms [95,96,97]. It is a progressive destruction of hard tooth tissues (enamel and dentine), involving demineralization of the inorganic parts as well dissolution of the organic components [98]. While enamel is almost a pure mineral (hydroxyapatite and derivatives) [99,100], mature dentine is composed of approximately 70% (in weight) inorganic components and 10% water [101]. The organic extracellular matrix constitutes 20%, of which 90% is collagen (mainly type I with minor components of types III and V); the remaining 10% is non-collagenous proteins such as DSP, DPP, DGP, dentin matrix protein-1 (DMP-1), and others [101]. Bacterial proteases of dental plaque as well as host-derived MMPs (and cathepsins) enable the breakdown of enamel and dentin; after degradation of the organic matrix, the capacity for remineralization is impaired [102]. A similar process may be observed in dental erosion; when the dissolution of enamel and dentin by acids (of non-bacterial origin) takes place, the organic matrix is exposed, and then the matrix can be degraded by host-derived MMPs, which block its remineralization [103,104]. Host-derived MMPs are synthesized during dentinogenesis and remain inactive in mineralized dentin, being activated by the acidic environment created during dentin demineralization [105]. Activation in an acidic environment can be caused by the induction of the cysteine switch as a consequence of prodomain conformational change or by the action of extracellular cathepsin proteases, which become active in the acidic environment [106]. When caries is left untreated, inflammation of the pulp proceeds (pulpitis) [65]. MMPs together with interleukins have a leading role in the initiation and development of pulp inflammation and may be responsible for its destruction [107], as the pulp ECM consists of collagens and non-collagenous proteins [108]. Due to bacterial infection within the root canal system, inflammation of the apical periodontium is observed (periapical periodontitis), which also involves the activity of MMPs; MMPs in periapical lesions are involved in a defensive reaction against bacteria and cause collagen degradation and the subsequent bone resorption [93,109]. 

MMPs, through degrading the ECM, are also involved in cancer invasion and metastasis [110,111,112,113]. Oral leukoplakia, which is the most common multifactorial premalignant lesion of the oral mucosa, may be induced by the inveterate use of tobacco in either smoked or smokeless form [114]; it has been shown that tobacco users have higher epithelial expression of MMPs, which imply their involvement in tobacco carcinogenesis [115]. High-risk human papillomavirus (HR-HPV) infection may be another cause of leukoplakia and carcinoma development [116,117,118]; HR-HPV E6 and E7 oncoproteins have an impact on cell homeostasis through inactivation of the p53 and pRB tumor suppressors, which leads to cellular immortalization [114], but it is not enough to explain the occurrence of cancer. Modification of the ECM by MMPs may represent another factor necessary for the establishment and progression of tumors [119]. It has been shown that the single nucleotide polymorphism (insertion of a guanine at position 1607 bp) in the promoter region of the gene encoding MMP-1 creates a core binding site for the transcription factors of the Ets family [120]; it has been associated with considerable up-regulation of MMP-1 transcription [120] and possibly increases the activity of this enzyme, which may promote HPV-induced carcinogenesis [119]. It has also been shown that this single nucleotide polymorphism may promote infection with HR-HPV [119]. 

MMP activity may be also associated with other oral diseases and problems like the failure of adhesive restorations over time [105]; oral lichen planus [63] and its transformation from reticular to erosive forms and further to carcinoma [121]; desquamative gingivitis associated with oral lichen planus [122]; odontogenic cysts and tumors (such as dentigerous cyst, odontogenic keratocyst, and unicystic ameloblastoma) [123]; as well as periodontitis.

## 5. The Extracellular Matrix of the Periodontium and the Role of Matrix Metalloproteinases in Periodontitis

### 5.1. The Anatomy of the Periodontium and Its Extracellular Matrix

The periodontium is a unique anatomical site, surrounding and supporting the teeth, anchoring them to the maxilla and mandible, and maintaining their functional and structural integrity during constant masticatory forces and bacterial presence. The periodontium is composed of a variety of connective tissues and includes four discrete principal components, namely, gingiva, periodontal ligament, cementum, and alveolar bone (Figure 4) [15]. 

The gingiva is a part of the oral mucosa and the outermost part of the periodontium, which consists of three principal layers—the epithelium with its BM, and the underlying lamina propria. The gingival epithelium (GE) can be divided into three regions: oral gingival epithelium (OGE), sulcular epithelium (SE), and junctional epithelium (JE) [124]. The OGE is the only keratinizing type of the GE, which covers the subepithelial connective tissue (the lamina propria) beyond the area of the gingival sulcus. The SE lines the gingival sulcus, apically bounded by the JE, which is a specialized GE attaching to the crown or root like a collar. The JE is critical to maintain the integrity of the periodontium and a key factor for the primary onset of periodontal diseases and treatments. The GE plays an important role as the first line of defense against invading bacteria [124,125,126,127]. The GE is associated with three distinct BMs. The OGE- and SE-BMs consist of typical components, including collagen IV, nidogen, and laminins (5, 7, 11, and LM-511). Adjacent to the tooth surface, the JE is sandwiched between two unique BMs—the external and internal basal laminae, consisting just of the lamina lucida and lamina densa. The external basal lamina is attached to the gingival lamina propria, whereas the internal lamina mediates attachment to the tooth surface. The composition of the external basal lamina is similar to that of OGE- and SE-BMs, but the internal basal lamina primarily comprises collagen VIII and laminin-5 [22]. The OGE and JE, compared to the SE, contain less E-cadherin [126], and the JE contains plenty of intercellular spaces across which neutrophils may migrate [128]. Proteoglycans found in the GE are CD44, syndecan-1, thrombomodulin, fibromodulin, lumican, and SPOCK1, and those found within its BM are perlecan, decorin (the proform), and biglycan [129]. Components of the GE-BM are synthesized by both the gingival epithelium and its underlying lamina propria [130].

The gingival lamina propria (gingival connective tissue) is a thin connective tissue layer underlying the GE and its BM, playing a major role in mechanical, nutritional, and immunological functions. The ECM of the gingival lamina propria is made up of collagen, elastin, proteoglycans, and non-collagenous structural glycoproteins. The gingival fibers are the major constituents of the gingival lamina propria, responsible for its mechanical properties and maintaining the framework and tone of the tissues, and primarily composed of type I and III collagens, as well as collagen V [15,33]. Collagen IV found in the lamina propria is associated with the vascular BM of the vessels present in the gingiva [131]. The gingival fibers form a highly complex system with fiber bundles arranged in various orientations [132]. The elastic system fibers, mainly produced by fibroblasts, comprise oxytalan, eluanin, and elastin, which are the second fibrillar proteins found in healthy gingiva, but in small amounts; oxytalan, elaunin, and elastin are distributed in the subepithelial, medium, and deep layers of the gingival lamina propria, respectively [15,33]. Oxytalan fibers are composed of fibrillin-containing microfibrils, while bundles of microfibrils intermingled with small amounts of elastin are composed of elaunin fibers [133,134,135]. Non-fibrous ECM components are other glycoproteins, glycosaminoglycans (but neither keratan sulphate nor chondroitin sulphate 6 were found in the gingival connective tissue), and corresponding proteoglycans, forming the gel in which the structural fibrous components and cells are embedded [15,33,129,136,137]. All glycosaminoglycans identified in human gingiva and the other ECM components are synthesized by gingival fibroblasts [137,138,139], with the exception of blood vessels. The gingival lamina propria contains also cellular components like fibroblasts, mastocytes, granulocytes, monocytes, macrophages, dendritic cells, lymphocytes, natural killer cells, and gingival mesenchymal stem cells; some of them are absent or not activated until microbial accumulation or invasion occur [15,140,141,142,143,144].

The periodontal ligament (PDL) is a unique, specialized, fibrous, and soft vascularized connective tissue responsible for anchoring firmly the teeth by the root cementum to the surrounding alveolar bone. The collagens are a major constituent of the PDL (except water), forming a dense and oriented network of fiber bundles embedded in intercellular substances, which in the bone and cementum form mineralized Sharpey’s fibers [145,146]. Collagen I (75%) and collagen III (20%) are the major collagen types of the PDL; type III collagen is covalently linked to type I collagen, which presents a relatively high abundance of hydroxyproline and cysteine, but low levels of hydroxylysine [147]. Other types found in the PDL are collagens IV, V, VI, XII, and XIV [148]. Elastic fibers, mostly composed of oxytalan fibers, appear to be a minor fibrous component of the PDL [15,135,149]. Other glycoproteins identified in the PDL are periostin, fibronectin, and tenascins [134,150,151,152]. Moreover, the PDL contains an abundance of glycosaminoglycans, and plenty of proteoglycans are identified in the PDL or PDL cells, such as CD44, syndecan-1 and -2, nerve/glial 2, perlecan, asporin/PLAP-1, decorin, biglycan, lumican, fibromodulin, and versican [129]. Fibroblasts are the major cell population within the PDL and are responsible for the synthesis and degradation of the ECM components [153]; compared with other fibroblasts, periodontal ligament fibroblasts are unique in that they possess the capacity to differentiate into cementoblasts and osteoblasts [152]. Other cell types include epithelial cell rests of Malassez, monocytes, macrophages, endothelial cells, and periodontal ligament stem cells (undifferentiated mesenchymal cells and neural crest stem cells, 95% and 5%, respectively). The latter represent a promising target for periodontal regeneration and a source for neuroregenerative applications (like spinal cord injury, Alzheimer’s disease, multiple sclerosis, and cerebral ischemia) [144,146,152,154].

The cementum is a bone-like mineralized (calcified) non-vascularized connective tissue layer covering the tooth root, involved in the attachment of the teeth to the surrounding alveolar bone by the PDL [155,156,157]. Human teeth cementum categories are based on the presence or absence of cementocytes in the structure (the acellular and cellular cementum), as well as the presence of collagen fibers (afibrillar and fibrillar) and their origin (intrinsic or extrinsic fibers, depending on their production by cementoblasts or by fibroblasts, respectively), which allows us to distinguish acellular afibrillar cementum (AAC), acellular extrinsic fiber cementum (AEFC), and cellular intrinsic fiber cementum (CIFC), which is usually present in combination with the AEFC as a component of mixed stratified cementum (MSC), with specific distribution along the length of the root [156,158]. Focusing on the cementum matrix composition, regardless of its histological kinds and their distribution, the content (by weight) of the inorganic (mineral) part of the cementum is around 65%, the organic content constitutes 25%, and the remaining 10% is water [159]. The inorganic part consists of hydroxyapatite crystals [160]. The major organic component is collagen, of which the predominant type is collagen I fibers, accounting for 90% of all collagens found in the cementum. Type I collagen serves as a reservoir for hydroxyapatite nucleation during cementogenesis, which successively develops into intrafibrillar apatite crystals [161]. A small quantity of collagen III, which coats collagen I fibers, is also found [161]. Cementum attachment protein is another collagenous protein [162]. Cementum contains many non-collagenous proteins, such as osteopontin, bone sialoprotein, matrix gamma-carboxyglutamic acid protein, osteocalcin, alkaline phosphatase, fibronectin, cementum-derived growing factor (CGF), cementum protein-1, and cementum enamel-associated proteins [161,163,164]. Glycosaminoglycan-containing proteoglycans seen in the cementum matrix are syndecan-1, syndecan-2, decorin, biglycan, fibromodulin, lumican, and versican [129,161,165,166]. The proteoglycans appear to inhibit collagen mineralization by occupying strategic locations intended to be filled with hydroxyapatite, hence their content in mineralized tissues is relatively low or absent; however, the initial phase of cementum formation requires proteoglycans accumulation, so they are intended to be degraded during cementogenesis [161]. The cementum is secreted by cementoblasts, fibers are produced by cementoblasts or fibroblasts, cementoblasts entrapped in the cementum, during cementogenesis, become cementocytes located inside the lacunae [167].

The alveolar bone (syn. alveolar process) is a vascularized structure supporting the teeth made of osseous (bone) tissue, a type of mineralized connective tissue. The alveolar bone structure consists of both cortical and trabecular bone. The cortical bone (syn. cortical plate) consists of plates of compact bone. The outer side of the alveolar process is composed of a typical cortical bone, while the bone directly surrounding the teeth (called the alveolar bone proper, bundle bone, or cribriform plate, radiographically termed the lamina dura) has plenty of holes, where Volkamann’s canals pass through and access the PDL. The alveolar bone proper contains Sharpey’s fibers, which are the mineralized endings of the PDL fiber bundles (the opposite endings are embedded in the cementum as extrinsic fibers, known under the same name). The alveolar crest is the most cervical part of the alveolar bone, where the outer and inner plates meet together, ending slightly apical to the cementoenamel junction. The cortical and cribriform plates are supported by the trabecular bone (syn. cancellous or spongy bone) located between them, which has less dense and more porous tissue, and is filled with marrow, providing vascular supply. The outer cortical plates are lined with the periosteum, with the gingiva forming the mucoperiosteum, while the inner cortical plates are lined with the PDL, both attached to the bone by Sharpey’s fibers [168,169]; trabeculae inside the trabecular bone are lined with the endosteum. Bone tissue has a composition of around (by weight) 65% inorganic parts, 25% organic matrix, and 10% water [170], which is similar to the cementum composition [159]. The bulk of the inorganic part is composed of hydroxyapatite, containing some impurities (substitutions of the phosphate group by carbonate, the calcium ions by potassium, magnesium, strontium, and sodium, and the hydroxyl groups by chloride and fluoride) [170,171]. The organic component of bone tissue comprises more than 30 proteins, with collagens being the most abundant—around 90% (the majority are collagen type I, but also types III and V) [171,172]. The remaining 10% are non-collagenous proteins which significantly contribute to bone biological function; among them are plenty of ECM proteins, as well as growth factors and cytokines, such as osteonectin, osteopontin, osteocalcin, bone sialoprotein, fibronectin, vitronectin, matrix Gla protein (MGP), bone morphogenetic proteins (BMPs), alkaline phosphatase, and thrombospondin, as well as albumin and alpha-2-HS-glycoprotein [173]. Among factors playing a role in bone remodeling are osteoprotegerin (OPG), receptor activator of nuclear factor kappa-Β ligand (RANKL), carbonic anhydrase II (CA II), cathepsin K, MMPs, and TIMPs [174,175,176,177,178]. Some proteins can be synthesized locally, whereas others are adsorbed from the circulation [173]. Embedded in Sharpey’s fibers in the bone are mineralized collagen fibers (predominantly composed of type I collagen, but also collagen III), associated with non-collagenous proteins like osteopontin and bone sialoprotein [179]. The periosteum consists of an outer layer, which is fibrillar, and an inner cellular layer; the outer layer is composed of collagens and elastin with fibroblasts, while the inner osteogenic layer contains osteoprogenitor cells (mesenchymal stem cells), which are precursors of osteoblasts, as well as mature osteoblasts, which are responsible for bone formation. Osteoblasts entrapped within the mineral matrix are called osteocytes, residing in the lacunae and canals called canaliculi, communicating with each other via cytoplasmic processes, and regulating bone remodeling; osteoclasts are a group of cells responsible for bone resorption. Within the bone marrow reside osteoprogenitor cells and other marrow cells [180,181,182,183,184,185].

### 5.2. A Brief Discussion of Periodontitis 

This article will not discuss the pathogenesis of periodontitis and associated conditions, on which there are many excellent papers [186,187,188,189,190,191,192,193,194,195,196,197,198,199], but we must present, in short, the most significant cellular and molecular changes affecting the periodontal ECM, causing anatomical disturbances. 

When the delicate balance between host response and bacterial virulence is disturbed, periopathogens found in dental plaque are able to advance gingivitis into periodontitis. Oral pathogens secrete a plethora of substances (e.g., lipopolysaccharides, gingipain proteases and other bacterial proteases, and leukotoxins), enabling them to invade a susceptible host. Across gingivitis and periodontitis stages, some cellular and intercellular changes are observed. Many host cells are involved in the beginning, progression, maintenance, and fluctuation of the inflammatory process; these comprise mast cells, granulocytes (commonly referred to as polymorphonuclear leukocytes, PMNs), natural killer cells, macrophages, fibroblasts, osteoclasts, endothelial cells, and epithelial cells. Plenty of substances playing an important role in inflammation are secreted by those cells, many of which are able to produce and secrete MMPs (i.e., fibroblasts, osteoblasts and osteoclasts, lymphocytes, granulocytes, macrophages, and keratinocytes) [65,200,201,202,203,204]. MMPs cause degradation of the ECM components, resulting in uncontrolled destruction of periodontal tissues, which is the most cardinal sign of periodontitis, including apical migration of JE, pocket formation, crest bone damage or loss, and subsequent consequences [136,186]. A schematic presentation of an inflamed periodontium and the most important features are presented in Figure 5 [140,143,205,206,207,208,209,210,211,212,213,214,215,216].

### 5.3. The Role of Matrix Metalloproteinases in the Periodontium and Periodontitis

As depicted in this review, collagens, non-collagenous proteins, and proteoglycans constitute the ECM, which is the scaffold of the entire body as well as the periodontium, and all are susceptible to the proteolytic activity of MMPs. If not snakes trying to kill and eat us, heavily mutilating us by its venom hyaluronidase and MMPs [217], there are plenty of conditions in which our own MMPs can severely affect our health. However, MMPs are vital in development and play a major role in the physiological ECM turnover. So, what is their role in the periodontium and periodontitis?

Turnover of the ECM, involving production and degradation, is a fundamental process of homeostasis [134]. The periodontal tissues normally exist in a steady-state equilibrium oscillating between tissue degradation and tissue neosynthesis [218]. The gingival epithelium turnover, with emphasis on the JE, is very fast [219]. Very little detailed information is available on the turnover of its BM in humans, but in the case of the glomerular BM, it seems to be rapid as well [220]. In marmoset, the remodeling of the gingival lamina propria is inordinately faster when compared to the other connective tissues [221]. The cementum does not undergo continuous remodeling, but continues to grow in thickness throughout life [160]. However, excessive mechanical forces (trauma), endodontic infections, or orthodontic forces may trigger cementum remodeling, not seen in regular conditions; Brochado Martins and colleagues even described a histologic and specialized remodeling structure of the cementum, which they referred to as the cementum remodeling compartment (CRC), resembling the bone remodeling compartment [222]. By contrast, the PDL and facing bone tissue are constantly being remodeled [223,224]. The PDL is continuously remodeled and the turnover rate in a rat model is twice as high as that of the gingiva, four times that of the skin, and six times that of the alveolar bone [225]. PDL fibroblasts are characterized by a high rate of collagen turnover [152,226], which is intensified by orthodontic forces [227]. Turnover in the alveolar bone has been reported to take place at a rate that is three to sixfold more robust than at non-oral skeletal sites, which implies that occlusal forces as well as oral biofilms modulate alveolar bone metabolism significantly [228].

In metabolism and remodeling of those tissues, plenty of factors are involved; according to our review, we focus on the direct role of MMPs. MMPs and TIMPs are involved in the remodeling of periodontal tissues, but their role is not crucial [229,230]. Proteins found in the BM, periodontal ligament, bone, and cementum are easily degraded by MMPs, but their activity is contained, resulting in an equilibrium between the synthesis and degradation of the ECM components. It is hypothesized that RECK also may play a role in the periodontium turnover [231].

In epithelial repair, keratinocyte migration is promoted and accelerated by MMPs, resulting in re-epithelialization, but the role of MMPs in this process does not rely solely on action on the matrix [232]. MMPs play an important role in the remodeling of cell surface molecules, such as the epidermal growth factor, other growth factors, and junctional proteins like E-cadherin. E-cadherin, the major constituent of the adherens junction, is a physiologic substrate of MMP-7, which mediates shedding (proteolytic release from the membrane) [233] or cleavage of E-cadherin [234], resulting in increased migration and proliferation of epithelial cells [235]. Produced in a healthy gingival epithelium, MMP-9 and MMP-2 promote the modification of cell adhesion to the BM by degrading collagen IV [232,236]. Gingiva connective tissue (lamina propria) turnover occurs through the extracellular MMP-dependent pathway as well as intracellularly by lysosomal enzymes (phagocytosis) [237], in which extracellular proteases (metallo and serine families) may be responsible for the initial phase of degradation. The intracellular route under physiological conditions seems to be more important [238,239,240]. MMPs and TIMPs are often observed at sites that histologically show signs of remodeling, like in healing adult human gingiva [238,239]. The turnover of gingival tissue may be disturbed as a side effect of specific drugs, such as immunosuppressants, calcium channel blockers, and anticonvulsants, resulting in gingival overgrowth caused by the inhibition of collagen degradation or enhanced collagen synthesis [9]. As Lauritano, Palmieri, and their colleagues presented in their study, gingival fibroblasts stimulated by cyclosporine A (a immunosuppressant) showed significant down-regulation for some MMPs, including the following genes: *Mmp8*, *Mmp11*, *Mmp15*, *Mmp16*, *Mmp24*, and *Mmp26*, while *Mmp12* and *Mmp13* presented up-regulation. In addition, the inhibition of collagen phagocytosis was observed, which can be a result of down-regulated integrins [241]. Both occurrences result in decreased collagen turnover and accumulation of the ECM in the gingival connective tissue, causing gingival overgrowth [241]. Also, nifedipine (a calcium channel blocker) is suggested to induce gingival overgrowth through the inhibition of collagen intracellular and extracellular degradation pathways by down-regulating MMPs [242], but the results are inconclusive in both cases. The PDL undergoes a constant physiological turnover, in which MMPs play a central role [243]. Under physiological conditions, collagen degradation in the PDL is carried out through phagocytosis by periodontal fibroblasts or extracellularly by MMPs synthesized by fibroblasts or PMNs. Once, it was thought that the extracellular breakdown of the PDL collagen induced by MMPs was associated with pathological conditions (gingivitis and periodontitis), but currently, it is recognized as a physiological process (especially during orthodontic tooth movement) [243]. Collagens in bone tissue are under constant turnover, and this process occurs through two different pathways, both associated with MMPs: the first involves ECM collagen degradation by secreted or membrane-anchored collagenolytic proteases; the second alternative path takes place intracellularly—MMPs are responsible for the pre-cleavage of collagen into soluble fragments through receptor-mediated (urokinase plasminogen activator receptor-associated protein, uPARAP/Endo180) uptake, which are endocytosed for further processes and delivered to the lysosomes where they are degraded by cathepsins (B, L, N, and K) under acidic conditions [177,244]. During orthodontic tooth movement, applied orthodontic forces cause extensive remodeling of the PDL and alveolar bone matrix [245], which, according to the cited systematic review [227], is mainly orchestrated by MMPs and TIMPs. It was reported that the inhibition of MMPs (by synthetic MMPs inhibitors) decreases orthodontic tooth movement by inhibiting the PDL and bone remodeling [245,246]. Presumably, in the cementum remodeling compartment, a similar process may occur to that observed in bone tissue.

Without a doubt, the degradation of collagen, for which MMPs are the key enzymes, is essential in the pathogenesis of periodontal diseases [136]. Host MMPs are assisted by bacterial collagenases, for instance, gingipain proteases secreted by *Porphyromonas gingivalis* or dentilisin and dentipain secreted by *Treponema denticola*, which are responsible for host proMMP activation [65,247,248]. Plenty of MMPs are involved in periodontitis. Across studies, the levels of expression and production of various MMPs are measured with the use of different detection techniques performed on different samples like saliva, mouth rinse, gingival crevicular fluid, peri-implant sulcular fluid, and excised tissue. Real-time polymerase chain reaction (RT PCR) and mRNA in situ hybridization (RNA ISH) are nowadays the most commonly adopted techniques for MMPs’ transcript detection. Protein immunoassays, such as enzyme-linked immunosorbent assay (ELISA), Western blotting, and immunohistochemistry (IHC), are widely used to detect the presence of MMPs but are limited in the detection of proteolytic activity, although various molecular forms of MMPs —proenzyme and active forms, single or together—can be assessed. Finally, there is not a clinically recognized single MMP biomarker, but frequently, MMP-8 is considered the most promising [249]. However, inflamed periodontal tissues reveal overall elevated levels of MMPs when compared to control healthy specimens. Uncontrolled MMP production and conversion of zymogenic forms into active forms in response to inflammation, as well as the inactivation of endogenous protease inhibitors, lead to an imbalance between MMPs and TIMPs, resulting in excessive degradation of periodontal tissue components.

The epithelium, in which the JE plays a strategic role, and the BM are the first and foremost barriers that protect the periodontal tissues from inflammation [231,250]. E-cadherin and the BM are substrates for host proteases (like MMPs) and bacterial proteases, and the following disintegration of epithelial cell–cell connections and the BM enables bacterial invasion into the underlying tissue, initiating and preserving inflammation. The initiation of periodontal disease is attributed to cleavage within the second or third cell layer directly attached to tooth cells in the coronal-most portion of the JE facing biofilms [251,252]. Following cleavage, the secretion of cytokines and chemokines causes the accumulation of PMNs in the JE, which results in disruption of the epithelium by neutrophil proteases (e.g., cathepsin G, neutrophil elastase, and MMPs) [251]. Expression of E-cadherin in inflamed gingival tissue is reduced by *P.gingivalis*-produced LPS, which translates into enhanced invasiveness of *P.gingivalis* into tissues [253]. *Aggregatibacter actinomycetemcomitans* also reduces levels of E-cadherin [254]. Increased epithelial permeability creates space for bacterial products’ diffusion and bacterial cells’ migration toward the BM and underlying connective tissue [255]. Studies have revealed several types of changes in the gingival BM during chronic periodontitis, like detachment from the basal cell, duplication or multilayer formation, dislocation, fragmentation, thinning, diminution, and complete disappearance [255,256].

In inflamed gingiva connective tissue, MMP expression is increased and is positively correlated with the severity of the disease; then, collagen fibers’ progressive loss and elastic fibers’ disappearance are observed, as stated in Ejeil and colleagues’ study [257]. In this study, MMP-1, MMP-9, and MMP-13 were analyzed and found to be increased in the results of dot blotting analysis; furthermore, gelatin zymography revealed higher levels of active forms of MMP-2 and MMP-9, which is particularly seen in severe gingival inflammation. This study also revealed no significant difference for the MMPs and TIMPs between the control group and the mild inflammation group in dot blotting analysis [257]. The authors concluded that several MMPs, particularly MMP-1, MMP-2, MMP-9, and MMP-13, are involved in gingival ECM degradation during periodontitis. Similar outcomes were revealed in a study by Kubota and colleagues [258]: in periodontitis-affected gingival tissue, the expression of MMP (MMP-1, MMP-3, and MMP-8) mRNA was higher in the diseased group than in the control group. However, Gonçalves and colleagues [259] revealed that MMP transcripts (for MMP-1, MMP-2, MMP-9, and MMP-13) found in gingiva samples were not significantly different in patients affected by gingivitis, chronic periodontitis, and aggressive periodontitis. Also, they revealed that MMP-2 mRNA was virtually absent in the periodontitis groups, but present in some gingivitis-affected tissues and healthy controls; MMP-9 transcripts were more frequently detected in periodontitis samples. In the semi-quantitative analysis, the expression of MMP-1 and MMP-13 was slightly higher in the disease-affected samples, MMP-9 was lower in the gingivitis-affected group, and—as said above—MMP-2 was nearly undetectable; all of the results were not statistically significant and highly heterogeneous. Moreover, in Western blotting analysis, MMP-2 was found in gingivitis and aggressive periodontitis samples and at very low levels in the chronic periodontitis and healthy controls groups; meanwhile, MMP-9 was abundant in gingivitis and aggressive periodontitis, less in chronic periodontitis-affected tissues, and absent in the control group. In the gelatin zymography assay, MMP-9 levels were higher in both periodontitis and gingivitis groups compared to control specimens, but with no statistical significance; MMP-2 activity was higher in gingivitis and aggressive periodontitis samples and lower in chronic periodontitis samples, both compared to the control group, with a statistically significant lower value for the chronic periodontitis group compared to the gingivitis group. Dahan and colleagues’ study [260] also did not reveal significant differences in the expression of mRNA-encoding MMPs in periodontitis-affected gingiva samples, presenting great heterogeneity among individuals. The lack of difference between periodontitis-affected and control samples may be related to the cyclical nature of periodontal attachment loss, which could be associated with sudden activation of latent enzymes rather than an increase in their synthesis [259]. Indeed, the increase in active MMPs mostly results from the activation of inactive enzymes released by neutrophil degranulation, induced by cytokines and bacterial virulence factors rather than by *de novo* synthesis [261,262]; during degranulation, MMPs are released into the extracellular space as zymogens, where they are activated by the cysteine switch mechanism [263]. As mentioned in Dahan and colleagues’ study [260], by employing in situ hybridization, MMP-1, MMP-2, and MT1-MMP mRNAs were detected in fibroblasts in the connective tissue adjacent to the SE, near inflammatory cells, which were negative for MMP transcripts. Fibroblasts, macrophages, and epithelial cells produce MMPs on demand, when PMN-type MMPs are produced during cell formation in the bone marrow, kept intracellularly in granules, and then released in an inflamed area; therefore, some MMP mRNA expression in the periodontal tissues may be invalid and not unbiased [258]. For example, MMP-8, a major periodontal disease-associated MMP, is mainly PMN-specific and its activity is orchestrated by cytokines and various bacterial virulence factors, causing PMN degranulation in the site of inflammation and not affecting PMN development (and PMN-type MMP-8 synthesis) in the bone marrow [264]. During gingivitis, there can be a slightly elevated MMP-8 level in gingival crevicular fluid, but in the latent form, during active phases of periodontitis, it is further elevated and mostly converted into the active form [264]. Some overall conditions may affect MMP synthesis or activity, as in the case of diabetes where an increased concentration of MMPs in periodontal tissues affected by chronic periodontitis is observed [265].

MMP-8 and MMP-9 are the most abundant MMPs in periodontal tissues reflecting periodontal disease severity, progression, and treatment response [3,266]. As stated above, MMP-8 (also known as collagenase 2 or neutrophil collagenase) is a major host-derived collagenolytic proteinase primarily responsible for the irreversible destruction of periodontal and peri-implant tissues, catalytically very efficient and able to break down almost all of the proteinaceous structural components of connective tissues and the BM [267]. MMP-8 is produced during neutrophil development in the bone marrow and stored as pro-MMP-8 in subcellular neutrophil-specific granules before being released and activated extracellularly at the sites of inflammation [268,269]. Indeed, most MMPs found in saliva originate from PMNs entering the oral cavity via the gingival sulcus [261]. MMP-8 production may be also induced by non-PMN-lineage cells like gingival and periodontal ligament fibroblasts [264]. A high concentration of MMP-8 in gingival crevicular fluid is linked to the severity of periodontitis, possibly decreasing after periodontal treatment, although maintaining considerable levels during persistent periodontitis [65]. In addition, the inhibition of MMP-8 has been shown to reduce periodontitis [269]. For such reasons, active MMP-8 (aMMP-8) is suggested as a more precise biomarker than total or latent MMP-8, MMP-9, MMP-2, MMP-3, MMP-13, MMP-7, MMP-1, calprotectin, myeloperoxidase, NE, TIMP-1, and bleeding on probing [269]. Active periodontal tissue destruction can be identified non-invasively in oral fluids (saliva, mouthrinse, gingival crevicular fluid, and peri-implant sulcular fluid) in correlation to aMMP-8 levels. One employed technique is a quantitative MMP-8 chairside/point-of-care oral fluid test, with a sensitivity of 76–90% and a specificity of 85-96% [269]. A chairside/point-of-care oral fluid test is an immunofluorescent or immunochromatographic assay employing different antibodies to detect both PMN and non-PMN-type MMP-8 isoforms, which may or not recognize different epitopes of the two isoforms [264,270]. This tool can be used for early disease detection as well as monitoring progression and evaluating therapeutic effects [271]. The current grading parameters allow us only to predict whether it is likely that the periodontal breakdown would occur in the future, but not the exact time when it would occur [272]. Sorsa and colleagues’ findings indicate that the aMMP-8 chairside/point-of-care mouthrinse detection can be implemented as a staging and grading biomarker in the Tonetti and colleagues’ periodontitis classification. This study group proposed a modified classification with implemented aMMP-8 for disease progression (grading) from mouthrinse samples (Table 3) [272,273]. 

The utility of mouthrinse measurements relies on the simplicity to collect and sample gingival crevicular fluid, instead of using filter papers or micropipettes [272]. Mouthrinse (or oral rinse) samples are collected by having patients rinse their mouth with a water-based medium or tap water, which is subsequently collected and analyzed [274,275]. Additionally, oral rinse samples significantly vary from saliva, as oral rinse is regarded as a fluid containing mainly the gingival crevicular fluid from all periodontal pockets, while saliva contains shed epithelial cells from oral mucous membranes, nasopharyngeal discharge, food debris, and bacteria and their products, and its content is also affected by gingival crevicular fluid flow from gingival pockets (in dentulous subjects) [274]. Similarities to Sorsa and colleagues’ findings [272] (for gingival crevicular fluid samples) can be found in Nędzi-Góra and colleagues’ study [276]; however, it is concluded that the total MMP-8 level appeared to be more reliable for grading than the aMMP-8 level, although outcomes are limited by a small sample size. According to Räisänen and colleagues [277], mouthrinse aMMP-8 measurements are more precise in comparison to saliva aMMP-8 measurements and seem the optimal way of measuring the active periodontal breakdown and predicting periodontal disease progression. Commercially available kits for aMMP-8 detection have been introduced into the market. Also, MMP-9 can be useful as a periodontal disease biomarker [257,278]. It was shown that MMP-9 presence is positively correlated with periodontitis severity, which decreases when treated [279]. The main source of MMP-9 is PMNs, detected during advanced periodontitis in the JE, the SE, and as a scattered deposit along connective tissues of periodontitis-affected gingival tissues, with a lower level in the epithelium not exposed to inflammation [280]. Also, MMP-9 may play a key role in bone resorption caused by osteoclasts, which is further discussed.

Alveolar bone loss, a hallmark of periodontitis, is a consequence of turnover disruption by ongoing inflammation, with increased osteoclast activity and resorption predominance over bone formation. Cathepsins (especially cathepsin K, CTSK) as well as MMPs play a role in bone tissue collagen breakdown. In the acidic environment created by osteoclasts, the dissolution of the mineral phase occurs, exposing collagen fibrils. At low pH, secreted by osteoclasts, cathepsins (acidic proteases) degrade native collagen, followed by ECM degradation by MMPs [281]. MMPs are active at neutral pH, but in the osteoclast resorption pit, a constant acidic pH is maintained, in which MMPs considerably retain their activity; for example, pro-MMP-9 was reported to be activated by CTSK under acidic conditions in osteoclasts [282]. CTSK was thought to play a dominant role in bone ECM degradation; however, it was found that in CTSK-null mice as well as CTSK-mutant humans, collagen degradation is continued, which is a result of a CTSK-independent collagenolytic system in osteoclasts, associated with MMP-9 and MMP-14, as Zhu and colleagues found in their study [283]. Additionally, they noted in a mouse model that MMP-9 and MMP-14 mRNA levels were up-regulated during macrophage-to-osteoclast transition to an extent far exceeding other MMPs, which correlated to increased bone density and protection from pathologic bone loss in double knock-out individuals. It was also demonstrated that *Ctsk−/− Mmp9−/−* mice had an osteopetrotic phenotype with a fivefold increase in bone volume [284]. MMPs are also necessary for the migration of precursor and immature osteoclasts to the bone surface through the BM [285,286]. In addition, Kim and colleagues’ study [287] revealed that MMP-9 is an epigenetic regulator—it enters the nuclear compartment where it cleaves the histone H3 N-terminal tail (H3NT), promoting osteoclastogenic gene activation (CTSK also may play a role as an epigenetic regulator of osteoclast gene expression [284]). It is suggested that bone resorption is also associated with gingipains, secreted by *Porphyromonas gingivalis*, with Arg-gingipain-A (RgpA, also known as gingipain R1) being one of them. In Wilensky, Polak, and colleagues’ study [288], RgpA was shown to suppress phagocytosis of leukocytes, to have a positive role in the recruitment of leukocytes, and to increase cytokine levels, which in turn may collectively contribute to increased alveolar bone resorption. On the other hand, gingipains are able to degrade human-β-defensin 3, affecting peptides’ antibacterial activity [289]. It was also shown that bone resorption was significantly reduced after infection with *rpgA* mutants in murine periodontitis models. Another gingipain, Lys-gingipain (Kgp, also known as gingipain K), was shown to degrade osteoprotegerin, an osteoclast inhibitory factor produced by osteoblasts, which is suggested to be a crucial factor for osteoclastogenesis and bone loss in periodontitis [290]. It was reported that gingipains degrade cytokines that suppress osteoclastogenesis (such as IL-4 and IFN-γ) [291]. Furthermore, *P.gingivalis* cells, cell extracts, spent media, or lipopolysaccharides stimulate the secretion of MMPs at a higher level than TIMPs in dendritic cells, PDL cells, gingival fibroblasts, and engineered human oral mucosa, and secreted MMP zymogens can be directly activated by gingipains [291]. *P.gingivalis* was also shown to inactivate α1-antitrypsin [292], α2-macroglobulin [292], elafin [293], α1-antichymotrypsin [294], and TIMP-1 [295] through proteolytic degradation [292,295]. Also, neutrophil apoptosis prevents the release of MMPs and other histotoxic contents of neutrophil granules, but *P. gingivalis* has the ability to delay this process by secreting antiapoptotic substances and up/down-regulating host anti/pro-apoptosis genes, resulting in the retention and accumulation of inflammatory cells and progressive periodontal destruction [296,297,298,299,300]. On the other hand, *P. gingivalis* induces apoptosis in human gingival epithelial cells via a gingipain-dependent mechanism [301].

## 6. A Few Words about MMP Inhibitors in the Therapy of Periodontitis as a Conclusion

When pathogens invade the host, the immune system recruits leukocytes to the site of infection and eradicates them. MMPs play multiple roles in the normal immune response to infection, for instance, to facilitate the migration of leukocytes to the site of invasion by degrading the ECM components and to modulate cytokine and chemokine production and activity, which drive immune cell recruitment [302]. But an excess of MMP activity and the dominance of MMPs over their inhibitors provide a meat chopper-like brutal and rapid degradation of the ECM (this wording was adopted from Guo and colleagues’ article title) [291,303]. Therefore, MMPs may represent an excellent therapeutic target, as also bacterial proteases may be. Some MMP inhibitors have applications in periodontitis management as adjuvants to mechanical debridement, such as subantimicrobial dose doxycycline (SDD), non-steroidal anti-inflammatory drugs (NSAIDs), bisphosphonates, hydroxamates (such as batimastat and marimastat), proanthocyanidin, and stannous fluoride [59,65,103,304,305]. Chlorhexidine, a popular antimicrobial locally applied disinfectant frequently used in periodontitis therapy, is a non-specific MMP inhibitor, acting via a cation-chelating mechanism [306,307]. Other ways to arrest periodontitis could be achieved by MMP activity neutralization using recombinant TIMPs or antibodies directed against MMPs; it was shown that antibodies against MMP-9 and MMP-14 in combination significantly inhibited bone resorption [283]. Bacterial proteases can be neutralized using gingipain inhibitors like antibodies [308]. Indeed, it has been shown that immunization with an anrgpA DNA vaccine (in mice) [309] or recombinant RgpA peptide [310], an RgpA-Kgp complex (in rats) [311], a combination of minor fimbriae protein, RgpA gingipain hemagglutinin domain 1, and RgpA gingipain hemagglutinin domain 2 (in mice) [312] protected the animals from alveolar bone loss after oral infection with *P.gingivalis*. 

## Figures and Tables

**Figure 1 ijms-25-02763-f001:**
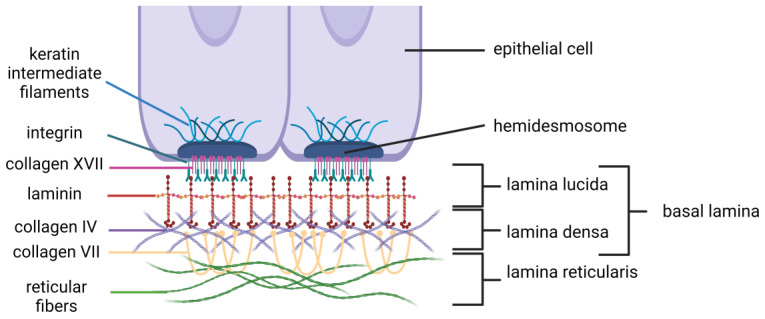
Schematic illustration of the basement membrane zones of epithelium. Description in the text. Created with BioRender.com.

**Figure 2 ijms-25-02763-f002:**
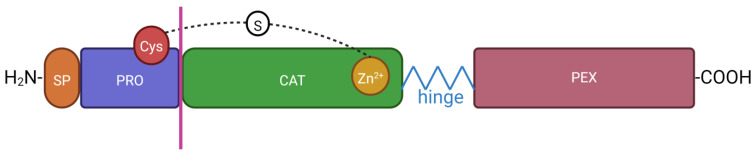
A schematic domain structure of MMPs. Abbreviations: (H_2_N–)—amino-terminus, SP—signal peptide, PRO—prodomain, purple vertical line—cleavage site, Cys—cysteine residue, CAT—catalytic domain, (–S–)—cysteine switch, Zn^2+^—zinc binding site, hinge—linker peptide, PEX—hemopexin-like domain, and (–COOH)—carboxyl-terminus. Created withBioRender.com.

**Figure 3 ijms-25-02763-f003:**
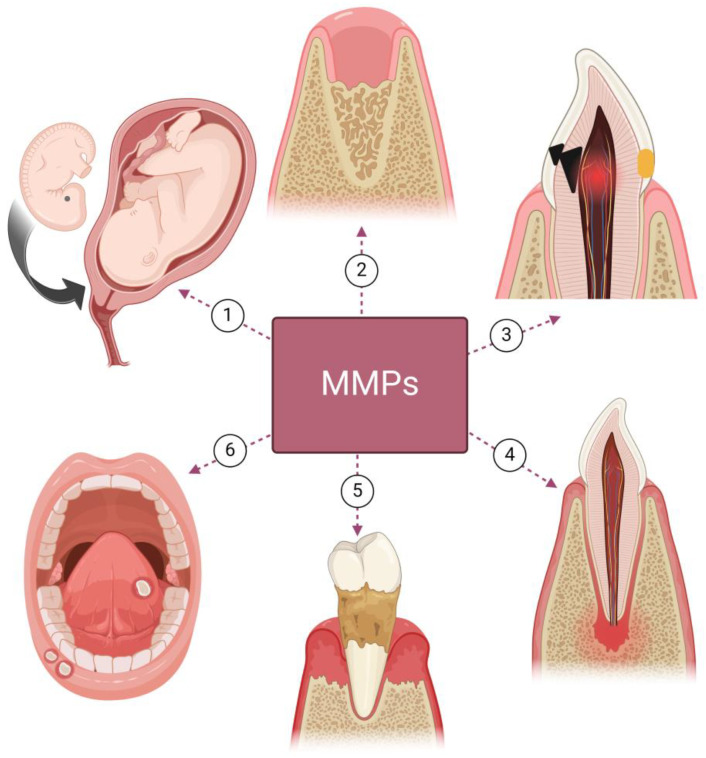
A schematic summary of MMPs’ role in the development and diseases of the oral cavity. MMPs are implicated in overall oral development (1) and wound healing like dental alveolus healing after tooth extraction (2). MMPs play a role in pathological processes, substantially contributing to the development of dental erosion, dental caries, pulpitis (3), periapical periodontitis (4), periodontitis (5), and oral lesions like lichen planus or squamous cell carcinoma (6). Created with BioRender.com.

**Figure 4 ijms-25-02763-f004:**
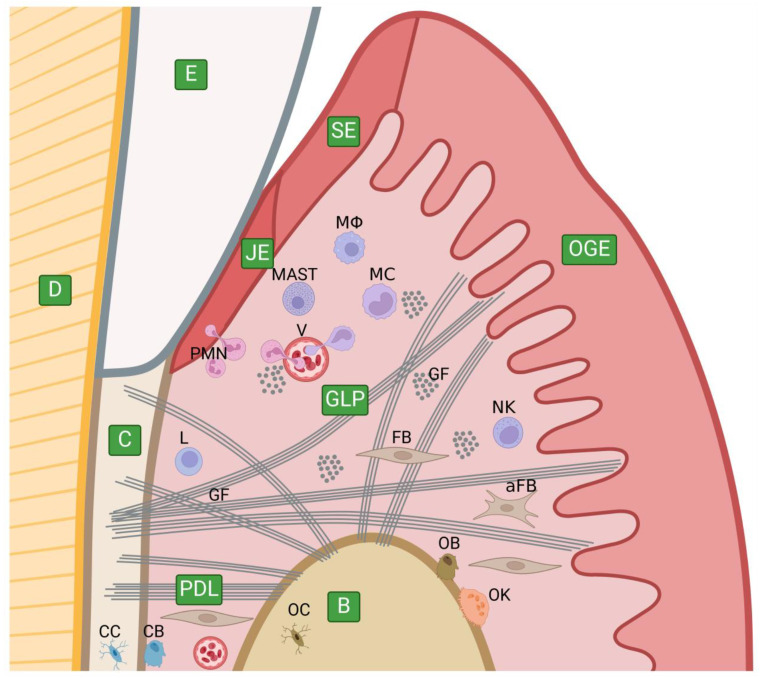
A schematic representation of the healthy periodontium. Abbreviations: 1. tissues: B—alveolar crest/bone, C—cementum, D—dentine, E—enamel, JE—junctional epithelium, SE—sulcular epithelium, OGE—oral gingival epithelium, GLP—gingival lamina propria, PDL—periodontal ligament, and V—blood vessel; 2. cells: MΦ—macrophage, MC—monocyte, MAST—mastocyte, PMN—polymorphonuclear leukocyte (granulocyte), (a)FB—(activated) fibroblast, NK—natural killer cell, L—lymphocyte, OB—osteoblast, OK—osteoclast, OC—osteocyte, CB—cementoblast, and CC—cementocyte; 3. GF—gingival fibers. Created with BioRender.com.

**Figure 5 ijms-25-02763-f005:**
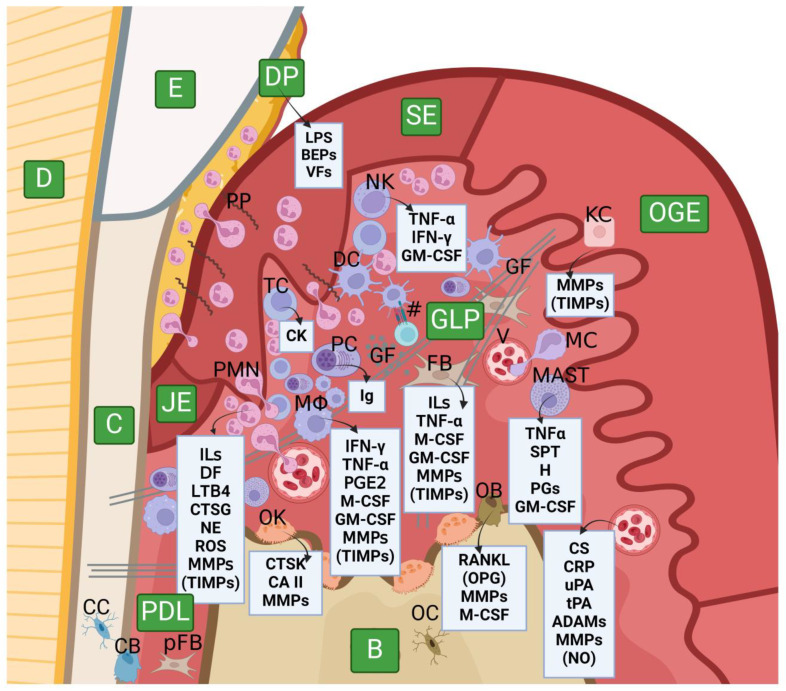
A schematic representation of the inflamed periodontium (periodontitis). Abbreviations: 1. tissues: B—alveolar crest (bone), C—cementum, D—dentine, DP—dental plaque, E—enamel, JE—junctional epithelium, SE—sulcular epithelium, OGE—oral gingival epithelium, GLP—gingival lamina propria, PDL—periodontal ligament, and V—blood vessel; 2. cells: DC—dendritic cell, #—CD4 T-cell and dendritic cell interaction (antigen presentation), MΦ—macrophage, MC—monocyte, MAST—mastocyte, PMN—polymorphonuclear leukocyte (granulocyte), FB—fibroblast, NK—natural killer cell, TC—T-cell, OB—osteoblast, OK—osteoclast, OC—osteocyte, CB—cementoblast, CC—cementocyte, PC—plasma cell, KC—keratinocyte, and PP—periodontal pathogens; 3. GF—gingival fibers; 4. substances: LPS—lipopolysaccharide, BEPs—bacterial extracellular proteinases, VFs—other virulence factors, e.g., gingipain proteases or leukotoxin, Ig—immunoglobulins, TNF-α—tumor necrosis factor alpha, IFN-γ—interferon alfa, GM-CSF—granulocyte-macrophage colony-stimulating factor, MMPs—matrix metalloproteinases, CK—cytokines, ILs—interleukins, M-CSF—macrophage colony-stimulating factor, PGE2—prostaglandin E_2_, LTB4—leukotriene B4, CTSG—cathepsin G, CTSK—cathepsin K, DF—defensins, NE—neutrophil elastase, ROS—reactive oxygen species, CA II—carbonic anhydrase II, RANKL—receptor activator of nuclear factor kappa-Β ligand, SPT—serine proteases, H—histamine, PG—proteoglycans, CS—complement system factors, uPA—urokinase-type plasminogen activator, tPA—tissue plasminogen activator, ADAMs—a disintegrin and metalloproteinase, (NO)—nitric oxide, (TIMPs)—tissue inhibitors of metalloproteinases, and (OPG)—osteoprotegerin (substances listed in brackets are involved in preventing tissue destruction, in contrast to some other listed substances). Created with BioRender.com.

**Table 1 ijms-25-02763-t001:** Classification of human matrix metalloproteinases based on substrate specificity [59,64,65].

Group	MMP	Name	Common Extracellular Matrix Substrates
Collagenases	MMP-1	collagenase 1, interstitial collagenase	collagen: I, II, III, VII, VIII, X, and XI; gelatin; pro-MMP-1, -2, and -9; fibronectin; laminin; tenascin; entactin/nidgoen
	MMP-8	collagenase 2, neutrophil collagenase	collagen: I, II, III, V, VII, VIII, and X; aggrecan; fibronectin; fibrinogen; gelatin; laminin; elastin
	MMP-13	collagenase 3	collagen: I, II, III, IV, VI, IX, X, and XIV; collagen telopeptides; gelatin; fibronectin; tenascin-C; aggrecan; fibrinogen; pro-MMP-9
Gelatinases	MMP-2	gelatinase A, 72kDa gelatinase	denatured collagens: I, II, III, IV, V, VII, X, and XI; aggrecan; elastin; fibronectin; gelatin (hydrolyzed collagen); laminin; entactin/nidgoen; pro-MMP-1, -2, -9, and -13
	MMP-9	gelatinase B, 92kDa gelatinase	denatured collagens: IV, V, XI, and XIV; aggrecan; gelatin (hydrolyzed collagen); elastin; fibronectin; laminin; dentin sialoprotein
Stromelysins	MMP-3	stromelysin 1	collagen: II, III, IV, V, VII, IX, X, and XI; collagen telopeptides; aggrecan; decorin; elastin; fibronectin; fibrinogen; gelatin; laminin; perlecan; entactin/nidgoen; versican; pro-MMP-1, -3, -7, -8, -9, and -13
	MMP-10	stromelysin 2	collagen: III, IV, V, VII, IX, X, and XI; collagen telopeptides; gelatin; elastin; fibronectin; laminin; aggrecan; decorin; perlecan; versican; fibrinogen; pro-MMP-1, -3, -7, -8, -9, and -13
	MMP-11	stromelysin 3	collagen: IV; gelatin; fibronectin
Matrilysins	MMP-7	matrilysin 1	collagen: I, IV, and X; aggrecan; elastin; fibronectin; gelatin; laminin; entactin/nidgoen; pro-MMP-1, -2, -7, and -9
	MMP-26	matrilysin 2	collagen: IV; gelatin; fibronectin; gelatin; pro-MMP-9
Membrane-type	MMP-14	MT1-MMP	collagen: I, II, and III; aggrecan; elastin; fibronectin; entactin/nidgoen; fibrinogen; gelatin; laminin; pro-MMP-2, -13, and -20
	MMP-15	MT2-MMP	collagen: I; aggrecan; fibronectin; gelatin; laminin; tenascin; pro-MMP-2
	MMP-16	MT3-MMP	collagen: I and III; gelatin; fibronectin; laminin; pro-MMP-2
	MMP-17	MT4-MMP	type I gelatin; fibrin; fibrinogen
	MMP-24	MT5-MMP	fibrin; gelatin; pro-MMP-2
	MMP-25	MT6-MMP	collagen: IV; gelatin; fibronectin; laminin; fibrin; fibrinogen; pro-MMP-2
Others	MMP-12	metalloelastase	collagen: I, IV, and V; aggrecan; elastin; gelatin; elastin; entactin/nidgoen; fibronectin; laminin; fibrinogen
	MMP-19	RASI-1	collagen: IV; fibronectin; gelatin; aggrecan; cartilage oligomeric matrix protein; laminin; fibrinogen
	MMP-20	enamelysin	collagen: IV and V; amelogenin; fibronectin; laminin; tenascin-C
	MMP-21	-	a1-antitrypsin
	MMP-23	cysteine array (CA)-MMP, femalysin	gelatin
	MMP-27	-	gelatin
	MMP-28	epilysin	unknown

**Table 2 ijms-25-02763-t002:** Classification of human matrix metalloproteinases based on their structure [60,61].

Group	Subgroup	MMPs
Secreted	Minimal Domain MMPs	MMP-7 and MMP-26
	Simple Hemopexin Domain-Containing MMPs	MMP-1, MMP-3, MMP-8, MMP-10, MMP-12, MMP-13, MMP-18, MMP-19, MMP-22, MMP-20 and MMP-27
	Gelatin-binding MMPs	MMP-2 and MMP-9
	Furin-activated MMPs	MMP-11 and MMP-28
	Vitronectin-like Insert Linker-less MMPs	MMP-21
Membrane-type	Transmembrane MMPs (with cytoplasmic domain)	MMP-14, MMP-15, MMP-16 and MMP-24
	Glycosylphosphatidylinositol (GPI)-anchored MMPs	MMP-17 and MMP-25
	Cysteine/Proline-Rich IL-1 Receptor-like Domain MMPs	MMP-23

**Table 3 ijms-25-02763-t003:** Proposed periodontitis grading classification with aMMP-8 implemented as biomarker; the use of high sensitivity CRP as well as aMMP-8 needs to be substantiated with specific evidence and validated. Primary criteria are not shown. Classification by Tonetti et al. [273] modified by Sorsa et al. (*) [272].

Periodontitis Grade			Rate of Progression		
			Grade A Slow	Grade B Moderate	Grade C Rapid
Grade modifiers	Risk factors	Smoking	Non-smoker	Smoker <10 cigarettes/day	Smoker ≥10 cigarettes/day
		Diabetes	Normoglycemic/no diagnosis of diabetes	HbA1c <7.0% in patients with diabetes	HbA1c ≥7.0% in patients with diabetes
Risk of systemic impact of periodontitis	Inflammatory burden	High sensitivity CRP	<1 mg/L	1 to 3 mg/L	>3 mg/L
(*) Biomarkers	Indicators of CAL/bone loss or collagen destruction	Mouthrinse and aMMP-8 level	<10 ng/mL (none) or 10 to 20 ng/mL (slow)	20 to 30 ng/mL	>30 ng/mL

HbA1c—glycated hemoglobin; CAL—clinical attachment loss; CRP—C-reactive protein; aMMP-8—active metalloproteinase 8.

## Data Availability

Not applicable.

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
