# Peer review of "Matrix Metalloproteinases in the Periodontium—Vital in Tissue Turnover and Unfortunate in Periodontitis"

_ijms, 2024, doi:10.3390/ijms25052763_

Round 1

Reviewer 1 Report

Comments and Suggestions for Authors

Dear authors, 

Thank you for the opportunity to review your manuscript on MMPs in periodontitis. I appreciate the effort you have put in to review the literature (296 references!), and the visual illustrations are quite nice.

In summary, I found the manuscript to be overly lengthy. It is not concise and the organisation can be improved. In its current form, I feel that this manuscript is more suitable as a textbook chapter, rather than a review paper. 

Narrative reviews typically involve an iterative and recursive process, concurrently analysing and interpreting information. Authors conducting such reviews need to provide examples supporting their interpretations, and illustrate how these interpretations contribute to their overall conclusions. I didn't see any of that in this manuscript, other than a lengthy summary of the known literature. No synthesis of new ideas from interpreting the existing literature was provided. Discussion on what is known is extensive and there is too much focus on this. The pertinent literature is missing. Things that I would consider more relevant about MMPs would be their clinical relevance, for example in the context of point-of-care testing and the strength of evidence as a biomarker.

To be published as a review, this paper needs extensive revision. To begin, I have the following suggestions:

1.     Remove ‘The review’ from the title. It is already clear that this is a review article.

2.     Typographical errors

a.     Line 13 ‘constituent’ should be ‘constituents’

b.     Lines 21-23 need to be rephrased. Suggestion: ‘During periodontitis, chronic inflammation and ECM destruction associated with the presence and activity of MMPs occurs, resulting in irreversible loss of periodontal tissues.’

c.     Lines 23-25 change ‘being’ to ‘is’, and add a comma before ‘as well as’

d.     Line 33 add a full stop after ‘which cells sit.’ Follow up with ‘It is composed of:’

e.     Line 47 remove ‘forming’

f.       Line 43 move the sentence ‘The ECM composition is unique…’ to line 48 just before ‘The ECM plays vital roles in…’

g.     Line 48 ‘provides’ should be ‘providing’

h.     Line 49 ‘constituents’

i.       Line 59 add ‘the’ in front of interstitial matrix

j.       Line 108 add ‘the’ in front of ECM. ‘Alter’ should be ‘alters’

k.     Line 109 add a comma in front of abundance. ‘Contribute’ should be ‘contributes’

3.     Remove ‘What is’ from the headings, this is unnecessary

4.     Lines 63 to 101. This section is unnecessarily wordy. The structure of the ECM has already been well-represented in Figure 1. Could the authors summarise this section, and then add it to the Figure description, which is currently too short. Summarise it in 150 words or less.

5.     The aim of the paper is not clear. Please clearly state the aim of the paper before the literature review sections. Structure the rest of the manuscript accordingly, and finish with a proper conclusion.

6.     With 296 references, this reads more like a textbook chapter rather than a review paper. Papers should be more concise and include only the key references. Sections 1 and 2 are too long for a background on MMPs.

7.     Section 3 is not brief. Perhaps the authors could use a figure to better illustrate how MMPs can affect various diseases of the oral cavity. It does not have to be complex. Put MMPs in the middle and have arrows branching out to the different conditions it can affect. The authors are already excellent at the visual representation of their figures, which I really appreciate.

8.      The sections are too long and include too much information that would be found in a textbook chapter.

9.     Line 505 ‘adding 16 references here is unnecessary. Please, just include the key references. There are key papers on the pathogenesis of periodontitis, together with the evolution of the paradigms on aetiology such as the following papers:

a.     https://pubmed.ncbi.nlm.nih.gov/9567963/

b.     https://pubmed.ncbi.nlm.nih.gov/26252398/

c.     https://pubmed.ncbi.nlm.nih.gov/32296429/

d.     https://pubmed.ncbi.nlm.nih.gov/33410172/

The key thing is that with advances in molecular sequencing techniques, our understanding on the pathogenesis of periodontitis has evolved. However, there are certain things that will always remain unchanged, such as a ‘susceptible host’ and ‘bacteria plaque challenge’, as demonstrated by Page and Schroeder in 1976.

10.  Please include more relevant information on MMP-8 and point of care testing. The authors talked a lot about MMPs. A general discussion is fine, but it would be good to zoom in on the key ones that are currently being used as biomarkers in clinical studies such as MMP-8 and 9. Please also include why these two in particular, have been primarily used as biomarkers of chronic periodontitis.

11. Table two is proposed by one group of authors and is not widely accepted. It is fine to include it as a guideline on the significance of MMP levels, but it is too premature to include it in the staging and grading criteria until it has been properly validated in other larger cohorts.

12.  Section 5 should be titled ‘Conclusion’ or ‘Summary’.

13. TIMPs and RECKs were mentioned in the manuscript as key regulators of MMPs. Are there any studies that have explored this as a therapeutic intervention to impede breakdown of tissues, as this seems to be the clinical relevance. The other relevance being used as biomarkers, if they are indeed accurate in estimating the extent of breakdown at a given point in time. 

14. The abstract reads like the introduction of this paper

Comments on the Quality of English Language

The English is generally fine, but there are a moderate amount of typographical errors so I suggest reviewing this again with a grammar checking software. 

Author Response

Dear Reviewer

We would like to thank you for your thorough review and valuable comments and suggestions. The manuscript has been revised accordingly. We believe that these changes have resulted in a greatly improved manuscript, which we hope is now suitable for publication in the International Journal of Molecular Sciences.

 Point-by-point response:

  1. ‘The review’ was removed from the title. 
  2. Thank you for these suggestions. They have been changed accordingly.
  3. Thank you for this suggestion. It has been changed accordingly.
  4. This section is a small summary to readers not acquainted with the field. Figure 1 does illustrate the basement membrane only, which is one part of the ECM, so we cannot remove the part concerning the interstitial matrix. To the figure description has been added ‘Description in the text’. 
  5. The aim of the study was added to a new section (1. introduction). We have also changed the title to reflect our objective appropriately 
  6.  Sections 1 and 2 are required for better understanding of the ECM and MMPs, usually avoided in such reviews. Indeed, as a background it is too long, so we have created a little introduction to state our objectives clearly. Also, according to IJMS instructions for authors, there is only a minimum word count and it counts 4000 words.
  7. We are pleased you appreciate our illustrations. A new figure has been added. 
  8. We agree. We are focused not only on MMPs roles, but on the periodontal ECM too. 
  9. Thank you for this suggestion. We have changed some of the references. We mention not only periodontitis pathogenesis, but also associated conditions, so other cited references we recognized as relevant. 
  10. More relevant information on MMP-9, MMP-8 and point of care testing have been added. 
  11. Indeed. CRP as well as aMMP-8 are not validated, so we have highlighted it as a proposal. 
  12. Thank you for this suggestion. 
  13. Indeed, we agree. But the review does not directly concern neither TIMPs, RECK nor treatment, but only mention them, and we would not expand this review. 
  14. The abstract is a summary of each section. 

Best regards

Dominik Radzki

Reviewer 2 Report

Comments and Suggestions for Authors

Thank you for the opportunity to review an article detailing the role of MMPs in periodontitis. Since the manuscript is in review format, the reader will find it easier to read if the style is appropriate.

Very detailed paper, however, there is little background statement as to why the author chose this topic. Also, the objectives are not clear. These should be clearly stated.

Materials and methods are unknown.

Criteria for selecting appropriate papers are not described.

Bias in the selection of reference was not considered.

The description of apoptosis, which is known so far, is not sufficient. An explanation concerning FasL is needed.

The reader wants to know if there are any new treatments that would control the activity of MMPs. Authors should add the latest findings on this topic.

Author Response

Dear Reviewer

We would like to thank you for your review. The manuscript has been revised accordingly.

Point-by-point response:

-        A background has been added

-        This review is neither a narrative review nor a systematic review, so there are not such requirements like materials and methods, risk of bias nor eligibility criteria

-        The review does not concern apoptosis, so we would like not to review the topic

-        The review does not concern any treatment, but focus on physiological and pathological roles of matrix metalloproteinases – in the summary authors mention some of the inhibitors 

Best regards

Dominik Radzki

Round 2

Reviewer 1 Report

Comments and Suggestions for Authors

4.       The special issue is titled ‘Periodontal disease, association with systemic conditions and pathogens.’ MMPs are neither a pathogen nor systemic condition, so I question if this journal is the right place for this paper. Consequently, the authors felt there was a need to overexplain things that were ‘not familiar’ to readers. It is not the purpose of reviews to educate people who are not from the same field.

5.       At nine lines, the introduction is way too short. There’s a lot of content in this paper. Surely there was a better way to write an appropriate introduction before stating the aims of the review.

6.       Your paper is at least 11000 words long. That is the length of an undergraduate’s dissertation, not a review article.

Title should only have the first word in upper case, same goes for the journal article titles in the references. My main issues with the paper are the purpose and its suitability for publication in this journal. The organisation of the paper is no good. For example, the aims and conclusion are not congruent.

Comments on the Quality of English Language

My main issues with the paper are the purpose and its suitability for publication in this journal. The organisation of the paper is no good. For example, the aims and conclusion are not congruent.

Author Response

Dear Reviewer

We would like to thank you for your thorough review and valuable comments and suggestions. The manuscript has been revised accordingly. We believe that these changes have resulted in a greatly improved manuscript, which we hope is now suitable for publication in the International Journal of Molecular Sciences.

 Point-by-point response:

  1. The article was transferred out of the Special Issue to regular submission under the Molecular Immunology section
  2. We state that the molecular part as well as periodontal part are relevant equally.
  3. The introduction has been changed accordingly.   
  4. According to IJMS instructions for authors, there is only a minimum word count and it counts 4000 words. Within last days have been published in IJMS a few papers counting more than 30 pages and more than 10 000 words (even 44 pages with 20 000 words). And each is a review article. For instance: 10.3390/ijms25031798, 10.3390/ijms25031826, 10.3390/ijms25031890, 10.3390/ijms25031845, 10.3390/ijms25031797. 
  5. The way of the title writing is according to ‘instruction for authors’ and Word Template, so not only the first word should be with a capital letter.  

Best regards

Dominik Radzki

Reviewer 2 Report

Comments and Suggestions for Authors

OK, but too long.

Author Response

Dear Reviewer

We would like to thank you for your review. 

According to IJMS instructions for authors, there is only a minimum word count and it counts 4000 words. Within last days have been published in IJMS a few papers counting more than 30 pages and more than 10 000 words (even 44 pages with 20 000 words). For instance: 10.3390/ijms25031798, 10.3390/ijms25031826, 10.3390/ijms25031888, 10.3390/ijms25031890, 10.3390/ijms25031845, 10.3390/ijms25031797

Best regards

Dominik Radzki

Round 3

Reviewer 1 Report

Comments and Suggestions for Authors

Dear authors, thank you for providing the revised version. I just have a couple more points to improve the manuscript:

1. The introduction is fine now, but I would appreciate insertion of the appropriate references. Currently there is not a single one.

2. The paper should also have a proper conclusion. 'A few words about MMPs inhibitors in therapy' is sufficient as a stand alone section.

Comments on the Quality of English Language

Nil

Author Response

Dear Reviewer

We would like to thank you for your thorough review.

  1. Appropriate references have been inserted. 
  2. Thank you for this suggestion. 
  3. Also, according to round 1 point 10, we have added a sentence: MMP-8 and MMP-9 are the most abundant MMPs in periodontal tissues reflecting periodontal disease severity, progression, and treatment response

Best regards

Dominik Radzki